# Death without Previous Hospital Readmission in Patients with Heart Failure with Reduced Ejection Fraction—A New Endpoint from Old Clinical Trials

**DOI:** 10.3390/jcm11195518

**Published:** 2022-09-21

**Authors:** Jacek T. Niedziela, Mariusz Gąsior

**Affiliations:** 13rd Department of Cardiology, Silesian Center for Heart Disease, 41-800 Zabrze, Poland; 23rd Department of Cardiology, Faculty of Medical Sciences in Zabrze, Medical University of Silesia, 40-752 Katowice, Poland

**Keywords:** heart failure, clinical trials, endpoints, methodology

## Abstract

Background: Most of the drugs approved and registered for use in heart failure (HF) therapy were examined in randomized clinical trials (RCTs) with the primary composite endpoint of death or hospital readmission. This study aimed to analyze the rates of the newly calculated event: death without prior hospital readmission, in HFrEF patients in large RCTs to show that the newly defined endpoint probably delivers additional data on the structure of the composite endpoint and helps to interpret the results of interventional studies. Methods: This study included RCTs on therapeutic interventions in HF patients. A literature search was performed, and 31 trials in which death without hospital admission could be calculated were included in the analyses. The death without a prior hospital admission endpoint was calculated as the difference between the composite endpoint rate (death or hospital readmission) and the readmission rate. The differences in the new endpoint between the study groups were calculated. Result: The death rates without prior hospital admission were lower in the intervention groups in five trials. In the SENIORS study, significant differences were found in the primary (composite) and death without previous hospital admission endpoints. In the ACCLAIM, VEST, and GISSI-HF STATIN trials, death without previous hospital admission was the only endpoint with a significant difference between the study groups. Moreover, the new endpoint rates were higher in the intervention group in the latter two studies. Conclusions: The new endpoint describing patients who died without prior hospital admission might be useful in previous and future interventional studies to provide additional data on the structure of the composite endpoint. Some therapies might reduce death without previous hospital admission rates, which could be beneficial, even without a reduction in overall long-term mortality.

## 1. Introduction

Poor outcomes characterize heart failure with reduced ejection fraction (HFrEF). The one-year all-cause mortality may exceed 20% while rates of one-year hospital readmission for heart failure and any cause reach 17% and 48%, respectively [1]. Modern drug and device therapies may improve prognosis, but even optimal medical treatment (OMT) does not provide satisfying results. Most of the drugs recommended for HFrEF patients were examined in large randomized clinical trials (RCTs), mainly with mortality and hospital readmission endpoints. The composite endpoint of mortality and hospital readmission assumes that both events worsen prognosis and should be avoided [2]. However, acute HF should be considered an urgent indication for hospital admission to provide adequate treatment and improve prognosis [3]. It was proven that an excessive reduction in hospital readmission might increase mortality in HFrEF patients [4]. For this reason, not all hospitalizations in HF should be avoided. The endpoint describing overall mortality rates includes patients who might have hospital readmissions before death. However, there is a group of HF patients who died without hospital admission in the whole follow-up. It may be calculated as a difference between the composite endpoint rate (death or hospital readmission) and hospital readmission rate. In our opinion, the results of some interventional trials have not been fully explored and described. Large clinical trials are designed to examine the primary endpoint. However, both positive and negative studies might require a more detailed outcomes analysis, as some therapies might reduce the death rate without previous hospital admission, which could be beneficial, even without reducing the overall long-term mortality. On the other hand, reducing hospital readmissions might not be beneficial when it is associated with high death rates without prior hospital admission. Therefore, we aimed to analyze the rates of the newly calculated event (death without prior hospital readmission) in HFrEF patients in large RCTs to show that the newly defined endpoint might deliver additional data on the structure of the composite endpoint and helps to interpret the results of interventional studies.

## 2. Methods

This study included RCTs on drugs, non-drug interventions, and device therapies in patients with HFrEF. As some trials were designed before the current definition of HFrEF, the inclusion criteria regarding the left ventricular ejection fraction (LVEF) were not the same. Thus, all trials with decreased LVEF, regardless of the cut-off threshold, were included.

A literature search was performed in September and October 2020 to select RCTs fulfilling the eligibility criteria. Two authors searched the MEDLINE database independently to screen titles and abstracts using the predefined protocol with the search query:

“(heart failure [Title]) AND ((death [Title/Abstract]) or (mortality [Title/Abstract]) or (composite endpoint [Title/Abstract]))”.

The search results included 19,157 studies evaluated according to the flowchart (Figure 1).

This study included RCTs that enrolled patients with HF/HF with decreased EF with drug or non-drug intervention or device therapy with the following endpoints: composite endpoint consisting of death or hospital admission and hospital admission; cause of hospital admission had to be the same as in the composite endpoint (e.g., all-cause death or hospital admission due to HF; hospital admission due to HF).

The analysis excluded trials with follow-up shorter than 30 days or trials discontinued early or suspended or with more than two parameters in the composite endpoint (in such a case, there is no possibility of calculating the rate of deaths without a rehospitalization event), or trials enrolling patients with asymptomatic left ventricular dysfunction instead of HF.

All disagreements were referred to as the third researcher, who made a final decision (and did not fulfill the criteria for being an author).

The full texts of all selected articles were obtained. The data was extracted to the five predefined templates: The list of included HF trials contains the trial name, study drug/device/intervention, year of publication, enrolment dates, number of participants, follow-up, cause of death, and cause of hospital admissionThe characteristics of patients included in the analyzed HF trials, including age, sex, HF etiology, HF duration, percent NYHA III/IV class, LVEF, heart rate, percent of implantable cardioverter-defibrillators (ICDs), cardiac resynchronization therapy with defibrillator (CRT-D), diuretics, beta-blockers, angiotensin-converting enzyme inhibitor (ACEI)/angiotensin receptor blockers (ARBs)/angiotensin receptor-neprilysin inhibitor (ARNI), and mineralocorticoid receptor antagonist (MRA) use.Tables with different endpoints’ combinations with trial name, study groups, follow-up, composite endpoint rates, death, and hospital admission. Based on the extracted data on events: death or hospitalization and death alone, the new endpoint—death without rehospitalization—was calculated as the difference between the two events (Figure 2). Depending on the combinations of the causes of death and hospital admissions, the following endpoints were calculated (Appendix A):
Death from CV causes without hospital admission for HF/worsening HF;Death for any reason without hospital admission;Death for any reason without CV hospital admission;Death for any reason without CV hospital admission;Death for any reason without any hospital admission;Death for CV reason without any hospital admission;Death for HF worsening without any hospital admission for HF worsening.


As no raw data is available regarding the mortality and follow-up (time to death), the difference between the death without hospital readmission rates was calculated using a 2 × 2 chi-square test (Appendix A). 

## 3. Results

The analysis included 31 trials, of which 18 were progressive-positive and 13 progressive-negative trials presented in Table 1. The baseline characteristics and treatment of patients in selected trials are shown in Appendix A.

The detailed rates of events are presented in Appendix A.

The number of trials and death rates without previous hospital admission, depending on their cause, are presented in Table 2. 

The detailed data on the endpoint rates in trials with a significant difference between groups regarding the death without hospital admission endpoint is presented in Table 3. In five trials, better outcomes in the intervention group were described using the new endpoint. In the PARADIGM-HF trial, the reduction of CV death without previous HF hospitalization was observed while in the MERIT-HF study, the intervention reduced all-cause death without prior HF hospitalization. In EPHESUS and SENIOR trials, a reduction in CV death without previous CV hospitalization was achieved. Moreover, in the EPHESUS study, a reduction in all-cause mortality without previous hospitalization for any reason was observed. In the ACCLAIM trial, death without previous hospital admission was the only endpoint with a significant difference between the study groups. In VEST and GISSI-HF STATIN trials, which did not show any essential differences between study groups in conventional endpoints, the death rates without previous hospital admission were higher in the intervention group (Table 3).

## 4. Discussion

Death without previous hospital admission is an easy to calculate endpoint in clinical trials as a difference between the rate of a composite endpoint of death or hospital admission and the rehospitalization rate. Depending on the cause of death and hospital admission in the composite endpoint, death rates without previous hospital admission may have different clinical interpretations and significance (Table 2). For example, deaths due to HF without HF hospitalization represent HF decompensation events without admission to the hospital. For the highest rates, greater hospitalization due to HF occurred, resulting in HF death. This parameter may indicate the quality of care and probably inappropriate HF treatment without needful hospital admission. The most prevalent (17 studies) composite endpoint of all-cause death without HF hospitalization represents all deaths without previous HF hospitalization. It also includes deaths not associated with HF or CV diseases, which may dilute the study intervention’s real effect on events. For that reason, it may have less clinical significance in terms of HF treatment effectiveness. However, a significant reduction in CV deaths is expected to influence all-cause mortality. Moreover, the difference between all-cause and cardiac-specific endpoints may be interpreted as a measure of significant adverse events [36].

Seven studies showed significant differences between the trial groups regarding death rates without prior hospital admission. In the PARADIGM-HF trial, a reduction in CV death without prior HF hospitalization was observed. Assuming that all acute HF cases were hospitalized, it may suggest that the trial drug reduced the number of CV deaths due to non-HF reasons or reduced the number of sudden cardiac deaths. In the MERIT-HF study, the intervention reduced all-cause death without prior HF hospitalization, suggesting the study drug’s positive impact on deaths other than HF-caused deaths. In EPHESUS and SENIOR trials, reduced CV mortality without previous CV hospitalization was achieved, which may describe the sudden cardiac death or death in the terminal phase of the disease when hospitalization might be intentionally avoided. Patients with CV diseases and a substantial risk of CV death are supposed to be hospitalized before death. In the SENIORS trial, the rates of CV death and separate hospital admission due to CV reasons did not differ between groups, and the only significant differences were found in the primary (composite) and death without previous hospital admission endpoints. Thus, the whole trial’s effect was probably driven by the differences in death rates without prior hospital admission.

In the ACCLAIM trial, death without previous hospital admission was the only endpoint with a significant difference between the study groups, as immune therapy reduced mortality without previous hospital admission with no effect on conventional endpoints. Using the new endpoint, the trial conclusions could influence future scientific directions regarding immune therapy in HF and possibly clinical recommendations. In another two trials without positive results in HF patients (VEST and GISSI-HF STATIN), death without previous hospital admission was the only endpoint with a significant difference between the study groups. Moreover, in both studies, the death rates without previous hospital admission were higher in the intervention group, meaning that patients in the placebo groups died more often in the hospital than in the intervention groups. It might be indirect proof of higher rates of sudden cardiac death in the intervention groups, as there is no reason for higher hospital mortality rates in patients in the placebo groups. It could also explain why the primary trial endpoints did not reach statistical significance (GISSI-STATIN and VEST trials).

Deaths without previous HF hospitalization may be a valuable clinical outcome and RCT event. However, contrary to hospital admissions due to myocardial infarction, not all HF hospitalizations are urgent. Hospital admissions due to HF may include acute (worsening) and non-acute hospitalizations (e.g., CIED implantations), leading to wrong conclusions. There is a clinical difference between hospitalization due to HF and worsening HF. Unfortunately, no different ICD-10 codes are available for chronic and acute HF provided for chronic and acute coronary syndromes. Thus, differentiation between urgent and non-urgent HF hospitalization may be difficult or impossible in retrospective studies. Moreover, in studies on myocardial infarction, non-fatal myocardial infarction was widely used as an endpoint and an element of a composite endpoint of death or non-fatal myocardial infarction. Such an endpoint probably cannot be used in HF, as it may be difficult to distinguish between a non-fatal acute HF and non-fatal chronic HF hospitalization.

In real-life settings, excessive reduction in HF rehospitalizations may increase mortality. Such an observation was made in the USA’s Medicare population, where hospitals with the worst hospital readmission rates were penalized. Consequently, the reduction in hospital admissions and economic profits was obscured by the higher mortality, as patients died outside the hospital [4]. Moreover, the endpoint “non-fatal HF hospital readmission” implies the assumption that fatal and non-fatal events are different regarding the clinical or pathophysiological mechanism [37].

On the other hand, the composite endpoint “death or non-fatal hospital readmission” favors mortality over rehospitalization as the composite endpoint’s element compared to the “death or hospital readmission”. According to the composite endpoint definition, the first registered event of a composite endpoint is considered the endpoint. In consequence, when death occurs during the hospital stay, the first endpoint (“death or non-fatal hospital readmission”) will be counted as death and the second (“death or hospital readmission”) as hospital readmission. The second issue is a time-to-event difference, which is supposed to be longer in the “death or non-fatal hospital readmission” endpoint, as patients who die in a hospital will be censored later. According to the Heart Failure Association of ESC statement, the preferred outcome in terms of mortality is cardiovascular mortality [36]. Among 30 analyzed trials, only 13 included CV mortality, which raises a question about the clinical interpretation and significance of trial outcomes.

In our opinion, the newly defined endpoint may have two practical applications. Our study showed that it could be easily calculated in most published trials. In such a case, it might explain the mechanism of death, as patients without rehospitalization before death most likely died of sudden cardiac death or were palliative patients who were not hospitalized and died at home. For that reason, it might help to interpret the outcomes of both positive and negative trials and potentially explain why some trials did not reach their goals in terms of the composite endpoints or hospital readmission. The conventional endpoints might not reveal all aspects of patient outcomes. Some therapies might reduce the death rate without previous hospital admission, which could be beneficial, even without reducing overall long-term mortality. Therefore, the second potential application of the newly defined endpoint may be in future clinical trials in different populations, including patients with heart failure, myocardial infarction, or after ICD/CRT-D implantation, to provide additional data on the structure of the composite endpoint. We do not intend to change or replace conventional hard endpoints with the newly defined endpoint as a primary endpoint but to deliver more detailed information on the outcome.

Our study has some limitations. Trials included in the analysis had different enrolment criteria and were conducted between 1986 and 2020, which affected individuals’ baseline characteristics, including drugs and device therapies (Appendix A). Moreover, follow-up varied between 1 and 57 months, which also influenced the rates of events. For these reasons, conducting a meta-analysis was methodologically impossible. The statistical methods for calculating the difference in death rates without previous hospital admission endpoints between groups were suboptimal. Unfortunately, the raw data on the trial events was unavailable, making it impossible to use optimal statistical tests. The most important limitation is that our retrospective calculation of the new endpoint rates cannot be interpreted as the prospective trial result, as all analyzed trials were designed for the endpoints used in their methodology, including calculating the power of statistical tests and sample size.

To sum up, the definition of endpoints in clinical trials plays a pivotal role in interpreting trial outcomes, affecting the clinical guidelines and recommendations. The new endpoint describing patients who died without prior hospital admission might be useful in previous and future interventional studies to provide additional data on the structure of the composite endpoint. Some therapies might reduce death without previous hospital admission rates, which could be beneficial, even without a reduction in overall long-term mortality.

## Figures and Tables

**Figure 1 jcm-11-05518-f001:**
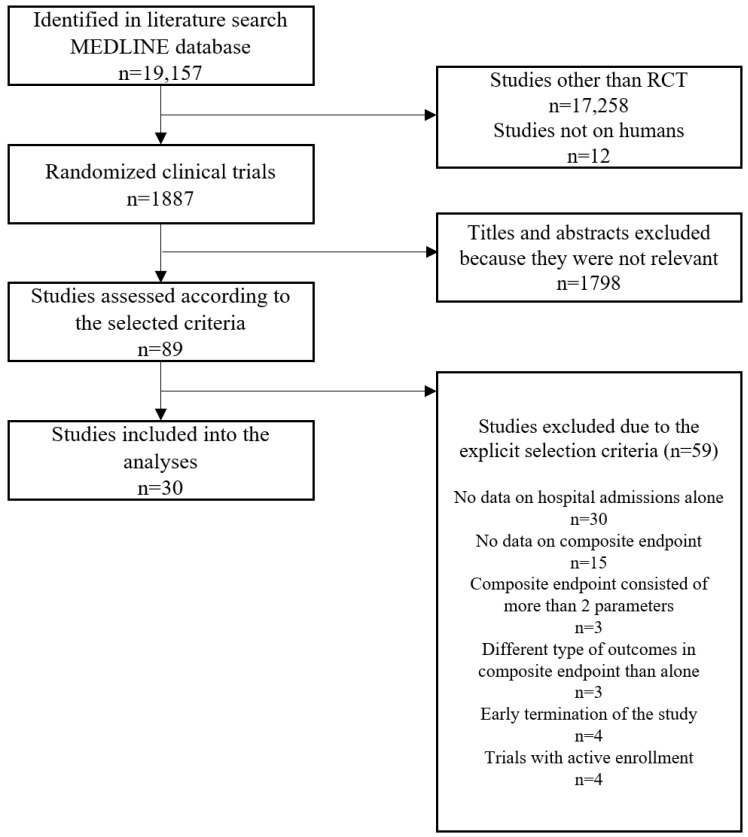
The summary of the evidence search and selection—flowchart. RCT–randomized clinical trial.

**Figure 2 jcm-11-05518-f002:**
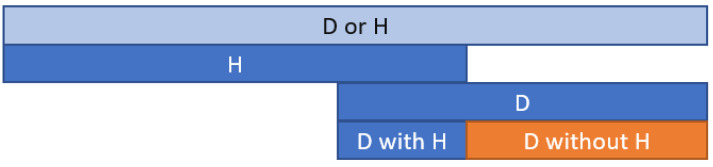
The death calculation method diagram without hospitalization (D without H) is based on the composite endpoint death or hospitalization (D or H). D—death; H—hospitalization.

**Table 1 jcm-11-05518-t001:** The characteristics of trials included in the analysis.

Trial	Drug/Device/Intervention	Year	Dates of Enrolment	Number of Patients	Follow-Up [Months]	Cause of Death	Cause of Admission
SOLVD-P [5]	ACEI	1992	1986–1990	4228	37.4	ALL	HF
DIG [6]	digoxin	1997	1991–1993	6800	37	HF	HF
MERIT-HF [7]	Beta-blocker	1999	1997–1998	3991	12	ALL	HF
CHARM-Alt [8]	ARB	2003	1999–2001	2028	33.7	CV	HF
CHARM-Add [9]	ARB	2003	1999	2548	41	CV	HF
EPHESUS [10]	Eplerenone	2003	1999–2001	6642	16	CV, ALL	CV, ALL
CHARM [11]	ARB	2004	1999–2001	4576	40	ALL, CV	HF
SENIORS [12]	Beta-blocker	2005	2000–2002	2128	21	CV	CV
CARE-HF [13]	ICD/CRT	2005	2001–2003	813	29.4	ALL	HF
HF-ACTION [14]	Exercise	2009	2003–2007	2331	30.1	ALL	ALL
HEAAL [15]	ARB	2009	2001–2005	3846	56.4 ^a^	ALL	HF, CV
MADIT-CRT [16]	ICD/CRT	2009	2004–2008	1820	54.0 ^a^	ALL	HF event
SHIFT [17]	Ivabradine	2010	2006–2009	6558	22.9	CV	HF
RAFT [18]	ICD/CRT	2010	2003–2009	1798	40	HF	HF
EMPHASIS-HF [19]	MRA	2011	2006–2010	2737	21	HF, CV, ALL	HF
PARADIGM-HF [20]	ARNI	2014	2009–2012	8399	27	CV	HF
DAPA-HF [21]	SGLT-2	2019	2017–2018	4744	18.2	CV	HF
EMPEROR [22]	SGLT-2	2020	2017–2019	3730	16	CV, ALL	HF
ELITE [23]	Losartan vs. Carvedilol	1997	1994–1995	722	11.2 ^a^	ALL	HF
VEST [24]	Inotrope	1998	1995–1996	3833	9.5 ^a^	ALL	HF
ELITE II [25]	ARB	2000	1997–1998	3152	18.5 ^a^	ALL	ALL
GISSI-HF [26]	PUFA	2008	2002–2005	6975	46.8 ^a^	CV	ALL
GISSI-HF [27]	Statin	2008	2002–2005	4631	46.8 ^a^	CV	ALL
ECHOS [28]	Anti-adrenergic	2008	2001–2004	1000	12	ALL	ALL
ACCLAIM [29]	Immune therapy	2008	2003–2005	2408	10.2	ALL	HF, CV, ALL
ASCEND-HF [30]	Nesiritide	2011	2007–2010	7007	1.0 ^a^	ALL	HF
ECHO-CRT [31]	CRT	2013	2008–2013	809	19.4	ALL	HF
RED-HF [32]	ESA	2013	2006–2012	2278	28	CV, ALL	HF
ASTRONAUT [33]	Aliskiren	2013	2009–2011	1639	11.3	CV	HF
ATMOSPHERE [34]	Aliskiren	2016	2009–2013	7016	36.6	CV	HF
COMMANDER HF [35]	Rivaroxaban	2018	2013–2017	5022	21.1	CV, ALL	HF

^a^—the follow-up in the study was presented in different units than months; ACEI—angiotensin-converting enzyme inhibitor; MRA—mineralocorticoid receptor antagonist; ARB—angiotensin receptor blockers; ICD—implantable cardioverter-defibrillator; CRT—cardiac resynchronization therapy; ARNI—angiotensin receptor-neprilysin inhibitor; SGLT-2—sodium-glucose co-transporter-2; PUFAs—polyunsaturated fatty acids; ESA—erythropoietin-stimulating agents; ALL–all-cause; CV–cardiovascular; HF–heart failure.

**Table 2 jcm-11-05518-t002:** Summary of the analyses of death without previous hospital admission.

Cause of Death	Reason for Hospital Admission	Death without Previous Hospital Admission			Number of Trials ^1^	Number of Significant Differences
		Lowest Rate [%]	Highest Rate [%]	Interpretation		
HF	HF	0.4	3.9	HF death without HF admission	2	0
CV	HF	6.2	15.2	CV death without HF admission	10	1
ALL	HF	3.2	21.9	All-cause death without HF admission	17	1 + 1 *
CV	CV	4.6	10.4	CV death without CV admission	2	2
ALL	CV	2.0	14.3	All-cause death without CV admission	4	1
CV	ALL	4.3	6.1	CV death without any admission	2	1 *
ALL	ALL	2.4	9.1	All-cause death without any admission	4	2

^1^—the number of trials in Table 2 is larger than the overall number of trials in this study, as, in some trials, more than one composite endpoint was evaluated; *—better in the placebo group. ALL–all-cause; CV–cardiovascular; HF–heart failure.

**Table 3 jcm-11-05518-t003:** Trials with a significant difference between groups regarding death without hospital.

Trial/Endpoint (Causes of Death/Hospital Readmission)	Group	n	Follow-Up [Months]	Composite Endpoint		Hospital Admission		Death		Death without Hospital Admission	
PARADIGM-HF	Sacubitril/Valsartan	4187	27	21.8%	<0.001	12.8%	<0.001	13.3%	<0.001	9.0%	0.004
CV/HF	Placebo	4212		26.5%		15.6%		16.5%		10.9%	
VEST	Placebo	1283	9.4	29.8%	0.25	18.5%	NS	18.9%	NS	11.3%	0.25
ALL/HF	Vesnarinone 30mg	1275		31.0%		18.2%		21.0%		12.8%	
ALL/HF	Vesnarinone 60mg	1275	9.5 ^2^	32.2%	0.63	17.0%	NS	22.9%	NS	15.1%	0.04 ^3^
MERIT-HF	Metoprolol CR/XL	1990	12	15.6%	<0.001	10.1%	0.004	10.8%	0.00009	5.6%	0.03
ALL/HF	Placebo	2001		21.9%		14.7%		7.3%		7.2%	
MERIT HF	Metoprolol CR/XL	1990	12	32.2%	<0.001	29.2%	<0.001	6.4%	0.00003	3.0%	0.002
ALL/ALL	Placebo	2001		38.3%		33.4%		10.1%		4.9%	
ACCLAIM	IMT	1204	10.2	33.1%	0.22	31.1%	0.39	9.7%	0.53	2.0%	<0.0001
ALL/CV	Placebo	1204		35.6%		29.6%		10.6%		6.1%	
SENIORS	Nebivolol	1067	21	28.6%	0.027	24.0%	0.2	11.5%	0.17	4.6%	0.019
CV/CV	Placebo	1061		33.0%		26.0%		13.7%		7.0%	
EPHESUS	Eplerenone	3319	16	26.7%	0.002	18.3%	0.09	12.3%	0.005	8.4%	0.0006
CV/CV	Placebo	3313		30.0%		19.6%		14.6%		10.4%	
EPHESUS	Eplerenone	3319	16	52.1%	0.02	45.0%	0.2	14.4%	0.008	7.1%	0.003
ALL/ALL	Placebo	3313		55.2%		46.1%		16.7%		9.1%	
GISSI-HF	Rosuvastatin	2285	46.8 ^1^	62.0%	0.409 ^4^	55.9%	0.613 ^4^	20.9%	0.804 ^4^	6.1%	0.007 ^3^
CV/ALL	Placebo	2289		60.5%		56.2%		21.3%		4.3%	

Admission endpoint. ^1^—different numbers of patients reached the endpoint; ^2^—follow-up was provided in other units than months; ^3^—better outcomes in the placebo group; ^4^—unadjusted; ALL–all-cause; CV–cardiovascular; HF–heart failure.

## Data Availability

The study reports data which is generally available.

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
