# Peer review of "Death without Previous Hospital Readmission in Patients with Heart Failure with Reduced Ejection Fraction—A New Endpoint from Old Clinical Trials"

_jcm, 2022, doi:10.3390/jcm11195518_

Round 1

Reviewer 1 Report

In summary, the authors define new endpoint for HF RCTs of death without prior hospitalization readmission. They included 31 trials in which death without hospital admission could be calculated and found that in 5 trials, a new endpoint showed better outcomes in the intervention group, and in 2 trials, the intervention was associated with worse mortality.

Introduction: consider being more succinct.

Methods: would consider changing thee inclusion/exclusion criteria from a list to paragraph form.

Results: would recommend that they present more of the relevant data included in the tables. They should consider highlighting the key take aways from the specific trials, and what further conclusions could be drawn.

Results: consider describing further the differences in using this endpoint in drug vs device RCTs.

Discussion: the authors should further describe the potential use of this novel endpoint, by illustrating what additional conclusions could be drawn from specific trials included in the analyses. 

Discussion: they should consider describing in what specific situations their new endpoint should be considered for future RCTs. They should further discuss the implications for drugs vs devices, etc. 

Discussion: consider including next steps for future work. Could they further analyze the patient populations with this new endpoint? Or are the trials to dissimilar?

Conclusion: in both the abstract and full manuscript, the conclusion described are rather vague, and I think thee authors should consider being more specific, and how it can impact further RCT design.

Author Response

In summary, the authors define new endpoint for HF RCTs of death without prior hospitalization readmission. They included 31 trials in which death without hospital admission could be calculated and found that in 5 trials, a new endpoint showed better outcomes in the intervention group, and in 2 trials, the intervention was associated with worse mortality.

Introduction: consider being more succinct.

The introduction was changed.

Methods: would consider changing thee inclusion/exclusion criteria from a list to paragraph form.

The inlcusio/exclusion criteria were rewritten.

Results: would recommend that they present more of the relevant data included in the tables. They should consider highlighting the key take aways from the specific trials, and what further conclusions could be drawn.

The tables in the manuscript were prepared to show important details of trials’ design. The conclusions that could be drawn in the specific trials regarding to the new endpoint are basically the same for all trials. Thus, we decided not to describe similar conclusions in each record.

Results: consider describing further the differences in using this endpoint in drug vs device RCTs.

Thank you for that suggestion. We have no idea what should be discussed in the drug vs. device RCTs. Probably the new endpoint will have the same meaning – reduction of the death rate.

Discussion: the authors should further describe the potential use of this novel endpoint, by illustrating what additional conclusions could be drawn from specific trials included in the analyses. 

We added the paragraph describing additional conclusion in some trials.

Discussion: they should consider describing in what specific situations their new endpoint should be considered for future RCTs. They should further discuss the implications for drugs vs devices, etc. 

The description was added.

Discussion: consider including next steps for future work. Could they further analyze the patient populations with this new endpoint? Or are the trials to dissimilar?

We added the paragraph in the discussion and in the conclusion sections.

Conclusion: in both the abstract and full manuscript, the conclusion described are rather vague, and I think thee authors should consider being more specific, and how it can impact further RCT design.

The conclusion section was rewritten.

Reviewer 2 Report

In this manuscript dr Niedziela and dr GÄ…sior, present their work on  “Death without previous hospital readmission in heart failure patients – a new endpoint from  old clinical trials”. I would like to congratulate them for the novelty of their work. However, I have some comments:

1.      In their introduction, authors present data about HFrEF treatment but the title does not specify whether their work refers to a specific subgroup of HF. Please clarify and correct concordantly.

2.      There is huge heterogeneity in the studies included in the analysis, both in terms of current treatment and sample. Their results cannot be further evaluated and they propose the endpoint Death without previous hospital readmission in heart failure patients to be used in further trials. Please comment.

3. In the results section the authors refer to the tables and comment the outcomes in trials. However further description is required. Please rewrite the section and clarify further.

Minor comments

Line 52: approved and registered. Please rephrase.

Line 169: please rephrase, the sentence doesn’t make sense.

Author Response

In this manuscript dr Niedziela and dr GÄ…sior, present their work on  “Death without previous hospital readmission in heart failure patients – a new endpoint from  old clinical trials”. I would like to congratulate them for the novelty of their work. However, I have some comments:

  1. In their introduction, authors present data about HFrEF treatment but the title does not specify whether their work refers to a specific subgroup of HF. Please clarify and correct concordantly.

The title was changed.

  1. There is huge heterogeneity in the studies included in the analysis, both in terms of current treatment and sample. Their results cannot be further evaluated and they propose the endpoint Death without previous hospital readmission in heart failure patients to be used in further trials. Please comment.

We added an appropriate comment.

  1. In the results section the authors refer to the tables and comment the outcomes in trials. However further description is required. Please rewrite the section and clarify further.

The results section was extended.

Minor comments

Line 52: approved and registered. Please rephrase.

The sentence was rephrased.

Line 169: please rephrase, the sentence doesn’t make sense.

The sentence was removed.

Reviewer 3 Report

In this study, Dr. Jacek T. Niedziela and colleagues investigated the new clinical endpoint of death rate without prior hospitalization using previous RCT studies. The unique ideas and perspectives are interesting to me. Conversely, as the authors have noticed, it doesn't seem easy to prove the scientific relevance when describing the authors' ideas.

Abstract

In the method heading, the explanation of the newly defined endpoint seems incorrect.

Introduction

I am not convinced by the logic described in lines 62-66. And the purpose of this study is unclear. Did you aim to clarify the importance of the newly defined endpoint by analyzing the rate of death prior to hospitalization? If you wanted to prove the clinical relevance of the new endpoint, you had to provide the information on how the clinical relevance of the new endpoint could be confirmed in the method section. As it is the current form, the manuscript just calculated the endpoint rates in the main text.

Methods

Please explain why you excluded the trials enrolling patients with asymptomatic left ventricular dysfunction.

Discussion

The setting of the primary endpoint is crucial for prospective interventional studies. The study size should be calculated using the event-occurrence rates previously proven in the other trials or a pilot study. Then, the primary endpoint cannot be freely determined to perform a realistically provable RCT study.

And the endpoint of all-cause death is essential for RCT. When an interventional study is planned, usually, the clinical effects for the target organ are already expected. In order to prove the clinical utility of a new therapy, the endpoint of all-cause death is warranted for investigating the benefits of taking the new treatment; whether it is greater than the real risks of the drug, including side effects. In this context, all-cause death and all-cause death or CV hospitalization are more reasonable for significant numbers of RCT than the death before the first hospitalization. You must explain the logic beyond the above in this manuscript through the introduction to the discussion.

Scheduled and chronic HF hospitalization are similar but not exactly the same. A scheduled hospitalization, including the CIED implantation, may be differentiated from worsening HF hospitalization by the definition of HF hospitalization in prospective studies.

As you mentioned, death prior to the first hospitalization may be meaningful in actual clinical practice, and it could be one of the causes of the failure to prove the new therapy's efficacy. After demonstrating the importance of the new endpoint, you should discuss the differences between the studies in accordance with the presence/absence of significant differences in the newly defined endpoint.

Probably I understand what you want to say and express through this study. However, as a scientific paper, the whole manuscript should be rebuilt under the clear considerations of what you want to prove, why you want to prove it, and how you prove it.

Author Response

In this study, Dr. Jacek T. Niedziela and colleagues investigated the new clinical endpoint of death rate without prior hospitalization using previous RCT studies. The unique ideas and perspectives are interesting to me. Conversely, as the authors have noticed, it doesn't seem easy to prove the scientific relevance when describing the authors' ideas.

Abstract

In the method heading, the explanation of the newly defined endpoint seems incorrect.

The explanation was rewritten.

  Introduction

I am not convinced by the logic described in lines 62-66. And the purpose of this study is unclear. Did you aim to clarify the importance of the newly defined endpoint by analyzing the rate of death prior to hospitalization? If you wanted to prove the clinical relevance of the new endpoint, you had to provide the information on how the clinical relevance of the new endpoint could be confirmed in the method section. As it is the current form, the manuscript just calculated the endpoint rates in the main text.

The introduction was modified.

  Methods

Please explain why you excluded the trials enrolling patients with asymptomatic left ventricular dysfunction.

Symptomatic and non-symptomatic patients with reduced EF are different populations with different recommendations. The current ESC guidelines are designed for HF patients.

Discussion

The setting of the primary endpoint is crucial for prospective interventional studies. The study size should be calculated using the event-occurrence rates previously proven in the other trials or a pilot study. Then, the primary endpoint cannot be freely determined to perform a realistically provable RCT study.

And the endpoint of all-cause death is essential for RCT. When an interventional study is planned, usually, the clinical effects for the target organ are already expected. In order to prove the clinical utility of a new therapy, the endpoint of all-cause death is warranted for investigating the benefits of taking the new treatment; whether it is greater than the real risks of the drug, including side effects. In this context, all-cause death and all-cause death or CV hospitalization are more reasonable for significant numbers of RCT than the death before the first hospitalization. You must explain the logic beyond the above in this manuscript through the introduction to the discussion.

The authors have no intention to replace the all-cause mortality or other endpoints with the new one. Our intention was to show that those endpoints might not reveal all aspects of patients’ outcome.

Scheduled and chronic HF hospitalization are similar but not exactly the same. A scheduled hospitalization, including the CIED implantation, may be differentiated from worsening HF hospitalization by the definition of HF hospitalization in prospective studies.

True. The senstences were changed.

 As you mentioned, death prior to the first hospitalization may be meaningful in actual clinical practice, and it could be one of the causes of the failure to prove the new therapy's efficacy. After demonstrating the importance of the new endpoint, you should discuss the differences between the studies in accordance with the presence/absence of significant differences in the newly defined endpoint.

The discussion was changed.

  Probably I understand what you want to say and express through this study. However, as a scientific paper, the whole manuscript should be rebuilt under the clear considerations of what you want to prove, why you want to prove it, and how you prove it.

We have changed the expression of the study. Hopefuly.

Round 2

Reviewer 2 Report

Congratulations to the authors for their effort. The topics pointed in my initial review were assessed.

Author Response

Congratulations to the authors for their effort. The topics pointed in my initial review were assessed.

Thank you very much.

Reviewer 3 Report

Thank you for your reply and revised manuscript. The purpose and relevance of your investigation become much more straightforward with a few added descriptions. I can know you understood what I wanted to tell you. Again, I am interested in the ideas of this study. But some issues stay unclear.

In the abstract, you describe, “The death without prior hospital admission endpoint was calculated as the difference between the rate of composite endpoint in the particular trial and the rate of “death” in the same trial.” In my understanding, your new endpoint was calculated as the difference between the composite endpoint rate and the readmission rate. Please verify again, and if your description is correct, please make a try to avoid the reader’s misunderstanding.

The fundamental purpose of this study was to show that the newly defined endpoint probably helps to interpret the results of interventional studies. It may be helpful in both positive and negative studies in considering why these results are achieved or why desired results are not gained. Is it right? The manuscript must correctly provide the author’s intention to readers, especially in the introduction section.

In the introduction, you should clarify why you focused on the new endpoint of death before admission. I am not convinced by the reason explained in the introduction section with its current description. For example, the questionable results of the previous interventional studies have not been discussed enough, and you guessed the newly defined endpoint might be helpful to explain them. Or did the evidence that an excessive reduction in readmission increase mortality give you a clue to consider the relevance of the death rate before the first admission? While it does not matter what the reason and route are, you should write the logical process for conducting this study based on your truth. The last sentence, “Some therapies might reduce death without previous hospital admission rate, which could be beneficial, even without a reduction in overall long-term mortality,” seems one of the significant aspects of this study, indicating the relevance of the new endpoint sharply. A clearer and more logical introduction that finally leads to the clear relevance of this study, like the above, is required.

The result section has improved and got easy for readers to figure out the results.

In the revised discussion section, balanced and enhanced discussions are provided now. The last paragraph in the limitation section seems essential. Similar to the last paragraph in the limitation, you should state that the authors do not intend to replace all-cause mortality or other hard endpoints with the newly defined endpoint as a primary endpoint. If you argue that the new one should be used as an endpoint and will have an interesting role in the future prospective study, you should state that clearly with your line of thinking with editing other parts of the manuscript. The purpose of the research must be consistent and clear.

I am looking forward to the next manuscript with better construction.

Finally, please explain your intentions in the reply file and indicate the place you edit in the text using page no. and line no. just after each explanation.

Author Response

Thank you for very valuable and constructive comments. I hope that our modifications are good enough to improve the structure and quality of the manuscript.

In the abstract, you describe, “The death without prior hospital admission endpoint was calculated as the difference between the rate of composite endpoint in the particular trial and the rate of “death” in the same trial.” In my understanding, your new endpoint was calculated as the difference between the composite endpoint rate and the readmission rate. Please verify again, and if your description is correct, please make a try to avoid the reader’s misunderstanding.

 Thank you, the sentence was changed. It was an embarrassing mistake.

The fundamental purpose of this study was to show that the newly defined endpoint probably helps to interpret the results of interventional studies. It may be helpful in both positive and negative studies in considering why these results are achieved or why desired results are not gained. Is it right? The manuscript must correctly provide the author’s intention to readers, especially in the introduction section.

 Thank you. It is a precious and constructive comment.

Changes were made in abstract (page 2, lines 29-31)

            In the introduction, you should clarify why you focused on the new endpoint of death before admission. I am not convinced by the reason explained in the introduction section with its current description. For example, the questionable results of the previous interventional studies have not been discussed enough, and you guessed the newly defined endpoint might be helpful to ex plain them. Or did the evidence that an excessive reduction in readmission increase mortality give you a clue to consider the relevance of the death rate before the first admission? While it does not matter what the reason and route are, you should write the logical process for conducting this study based on your truth. The last sentence, “Some therapies might reduce death without previous hospital admission rate, which could be beneficial, even without a reduction in overall long-term mortality,” seems one of the significant aspects of this study, indicating the relevance of the new endpoint sharply. A clearer and more logical introduction that finally leads to the clear relevance of this study, like the above, is required.

Changes were made - introduction page 4, lines 83-93.

The result section has improved and got easy for readers to figure out the results.

In the revised discussion section, balanced and enhanced discussions are provided now. The last paragraph in the limitation section seems essential. Similar to the last paragraph in the limitation, you should state that the authors do not intend to replace all-cause mortality or other hard endpoints with the newly defined endpoint as a primary endpoint. If you argue that the new one should be used as an endpoint and will have an interesting role in the future prospective study, you should state that clearly with your line of thinking with editing other parts of the manuscript. The purpose of the research must be consistent and clear.

We have changed paragraphs in the discussion (page 12, lines 260-275 and page 14 lines 289-293.